# Acute, Low-Dose Neutron Exposures Adversely Impact Central Nervous System Function

**DOI:** 10.3390/ijms22169020

**Published:** 2021-08-21

**Authors:** Peter M. Klein, Yasaman Alaghband, Ngoc-Lien Doan, Ning Ru, Olivia G. G. Drayson, Janet E. Baulch, Enikö A. Kramár, Marcelo A. Wood, Ivan Soltesz, Charles L. Limoli

**Affiliations:** 1Department of Neurosurgery, Stanford University, Stanford, CA 94305, USA; kleinp@stanford.edu (P.M.K.); isoltesz@stanford.edu (I.S.); 2Department of Radiation Oncology, University of California, Irvine, CA 92697, USA; yalaghba@uci.edu (Y.A.); ngoclied@uci.edu (N.-L.D.); nru@uci.edu (N.R.); odrayson@uci.edu (O.G.G.D.); jbaulch@hs.uci.edu (J.E.B.); 3Department of Neurobiology and Behavior, University of California, Irvine, CA 92697, USA; ekramar@uci.edu (E.A.K.); mwood@uci.edu (M.A.W.)

**Keywords:** cognitive dysfunction, electrophysiology, long-term potentiation, neutrons, space radiation

## Abstract

A recognized risk of long-duration space travel arises from the elevated exposure astronauts face from galactic cosmic radiation (GCR), which is composed of a diverse array of energetic particles. There is now abundant evidence that exposures to many different charged particle GCR components within acute time frames are sufficient to induce central nervous system deficits that span from the molecular to the whole animal behavioral scale. Enhanced spacecraft shielding can lessen exposures to charged particle GCR components, but may conversely elevate neutron radiation levels. We previously observed that space-relevant neutron radiation doses, chronically delivered at dose-rates expected during planned human exploratory missions, can disrupt hippocampal neuronal excitability, perturb network long-term potentiation and negatively impact cognitive behavior. We have now determined that acute exposures to similar low doses (18 cGy) of neutron radiation can also lead to suppressed hippocampal synaptic signaling, as well as decreased learning and memory performance in male mice. Our results demonstrate that similar nervous system hazards arise from neutron irradiation regardless of the exposure time course. While not always in an identical manner, neutron irradiation disrupts many of the same central nervous system elements as acute charged particle GCR exposures. The risks arising from neutron irradiation are therefore important to consider when determining the overall hazards astronauts will face from the space radiation environment.

## 1. Introduction

As humans undertake long-duration space exploration beyond low Earth orbit, including missions to the Moon and Mars, they will undergo exposures to a variety of high energy particles. These include galactic cosmic radiation (GCR) exposures that present a potential hazard to astronauts, likely increasing risks of both carcinogenesis and central nervous system disruptions [1,2]. Some degree of GCR exposure to astronauts during missions currently remains unavoidable, because providing sufficient spacecraft shielding to substantially minimize high energy GCR particle penetration would prohibitively increase spacecraft mass [3,4,5]. Thus, there is a critical need to thoroughly understand the risks of GCR irradiation on central nervous system function.

Past studies have well established that cognitive deficits arise following acute irradiation at mission-relevant doses (<0.5 Gy) [1] with protons [6,7,8] and ^4^He [9,10,11], the most prevalent GCR components. Cognition also becomes negatively impacted by less abundant high atomic number, high energy (HZE), fully ionized nuclei GCR components including ^16^O [12,13,14], ^28^Si [15,16], ^56^Fe [17,18,19] and some combined 2–3 ion exposures [8,20,21,22,23]. Furthermore, we have previously demonstrated that low dose irradiation is sufficient to perturb neuronal intrinsic, synaptic and network properties, whether in response to protons [24,25], ^4^He [9] or some of the most advanced GCR simulation yet with 5 combined ions [26].

However, considerably less is understood about how the non-charged components of the space GCR environment impact nervous system function. While neutrons occur with limited fluence in free space, a considerable proportion of the effective radiation field within a spacecraft could potentially be composed of neutrons (≈10–30%), as HZE ions are blocked by the shielding material and albedo neutrons are generated [2,27,28]. Fully understanding the relative risks of neutron irradiation is essential, because shielding strategies focused on reducing HZE exposures may lead to counterproductive elevations in the potentially more hazardous neutron radiation field within the spacecraft [29,30]. Unlike direct impacts from charged particles, neutron irradiation primarily causes tissue damage through the generation of recoil protons [31,32]. Despite differences in the exact energy deposition patterns of charged particles and neutrons, both are elements of the space radiation environment that pose a potential risk to astronaut nervous system function. An advantage of neutron exposure studies over accelerator-based GCR simulations is that more realistic radiation dose-rates of around 0.5–1 mGy per day [1,33] can be simulated during chronic exposures lasting several months using ^252^Cf sources [34]. We discovered that chronic, low dose (18 cGy) neutron irradiation at realistic low dose-rates resulted in suppressed hippocampal neuronal excitability, as well as perturbed hippocampal and cortical long-term potentiation [35]. Such underlying neurological disruptions were associated with mice displaying severe learning and memory impairments, and elevated anxiety-like behaviors. However, that study was unable to provide a more direct comparison of whether neutrons cause a similar range of nervous system disruptions within the acute exposure timeframes that have been used in the vast majority of charged particle GCR irradiation studies.

Our current study is therefore designed to broadly examine the neurological impacts of acute neutron irradiation. Using a range of approaches, we determined that exposure to space-relevant doses of acute neutron radiation disrupts the functional properties of individual neurons, alters synaptic signaling activity within neuronal networks and impairs memory capabilities within mice. Improved understanding of the central nervous system risks of space radiation exposures is essential to appropriately gauging the hazards posed to astronauts during long-duration spaceflight and developing appropriate measures to safeguard crew health during future missions.

## 2. Results

To probe the impact of acute neutron irradiation on the central nervous system, adult male mice received a single 18 cGy dose of whole body neutron irradiation, equivalent to an expected dose received by astronauts in transit to Mars [3,33]. Mirroring our prior assessment of chronic neutron irradiation [35], we then tracked alterations to multiple levels of nervous system that persisted for months beyond the initial insult using whole cell electrophysiology, local field potential recordings and a broad panel of behavioral assays.

### 2.1. Acute Neutron Irradiation Alters Excitatory Hippocampal Neuronal Activity

Past studies have repeatedly demonstrated that acute exposures to charged particle cosmic radiation induces significant alterations within the hippocampus, a critical structure for learning and memory processes. These include disrupted neuronal morphology [12,36], intrinsic excitability [24] and synaptic signaling properties [24,25,26] of hippocampal neurons. Additionally, chronic low dose neutron irradiation reduces hippocampal neuron excitability and diminishes excitatory synaptic inputs [35]. We therefore initially investigated whether acute, low dose neutron radiation posed a similar hazard to the cellular-level properties of hippocampal function in mice. To account for the nested data produced when multiple whole cell electrophysiology recordings were performed in the same animal, differences between treatment groups were evaluated by a linear mixed-effect model regression (LMM) analysis [37]. To supplement our statistical inference analysis derived *p*-values, we also include estimation statistics-based confidence intervals [38,39].

We first assessed whether acute exposure to neutrons at a space-relevant dose of 18 cGy persistently impacted the intrinsic electrophysiological properties of hippocampal pyramidal neurons within the CA1 superficial layer at 3–5 months following irradiation (Figure 1). Acute neutron irradiation did not alter the resting membrane potential of CA1 neurons (Mean difference (*M*_diff_) = 3.68 mV, 95% CI [−1.10, 8.75]; *d* = 0.54, 95% CI [−0.25, 1.32]; Linear mixed-effect modeling z-value (LMM *z*) = 1.42, *p* = 0.156; Figure 1A). We then applied a range of brief current injections to the CA1 pyramidal neurons from neutron-irradiated and control mice to test for changes in cell-intrinsic properties (Figure 1B). Acute neutron irradiation neither altered CA1 pyramidal neuron input resistance (*M*_diff_ = 73.8 MΩ, 95% CI [−22.1, 206.7]; *d* = 0.47, 95% CI [−0.29, 1.11]; LMM *z* = 1.11, *p* = 0.268; Figure 1C), nor hyperpolarization sag amplitude when neurons were injected with a −100 pA current (*M*_diff_ = 1.32 mV, 95% CI [−0.99, 5.67]; *d* = 0.31, 95% CI [−0.47, 1.00]; LMM *z* = 0.92, *p* = 0.357; Figure 1D). The threshold for CA1 pyramidal neuron activation was not altered by acute neutron irradiation, with an equivalent rheobase current evoking action potentials in irradiated and control neurons (*M*_diff_ = −5.80 mV, 95% CI [−28.6, 22.5]; *d* = −0.17, 95% CI [−1.04, 0.64]; LMM *z* = 0.43, *p* = 0.667; Figure 1B,E). Likewise, the absolute voltage threshold for action potential initiation was unchanged (*M*_diff_ = −1.37 mV, 95% CI [−4.68, 1.98]; *d* = −0.29, 95% CI [−1.05, 0.50]; LMM *z* = 0.74, *p* = 0.461; Figure 1F), along with other action potential characteristics (Appendix A). However, examining the full range of action potential frequencies evoked by current injections varying from 0 to 300 pA, acutely neutron-irradiated neurons displayed elevated responses (F_(1,695)_ = 70.9, *p* < 0.001, two-way ANOVA; Figure 1G). Altogether, while not altering certain intrinsic properties, acute neutron irradiation does elevate the output signaling responses of CA1 pyramidal neurons.

Charged particle GCR exposures are known to disrupt hippocampal synaptic markers, dendritic spines [12,36] and synaptic signaling properties [24,25,26]. Furthermore, we also observe that chronic neutron irradiation suppresses the frequency of excitatory synaptic inputs to CA1 pyramidal neurons [35]. Therefore, we next performed electrophysiological recordings of the spontaneous excitatory and inhibitory postsynaptic activity received by CA1 pyramidal neurons to assess whether acute neutron irradiation alters hippocampal connectivity (Figure 2). Consistent with chronic exposures, we detected a large effect-size decline in the spontaneous excitatory postsynaptic current (sEPSC) frequency received by CA1 pyramidal neurons following acute neutron irradiation (*M*_diff_ = −2.20 Hz, 95% CI [−4.04, −0.47]; *d* = −0.87, 95% CI [−1.58, −0.03]; LMM *z* = 2.13, *p* = 0.033; Figure 2A,B). To assess potential differences in the sEPSC characteristics of individual neurons, all sEPSCs detected within a 200 s recording period from each cell were averaged together to generate a standard profile (Figure 2C). Although there was some variability among neurons, acute neutron irradiation had no overall impact on average sEPSC amplitude (*M*_diff_ = −2.71 pA, 95% CI [−5.43, 0.0]; *d* = −0.72, 95% CI [−1.43, 0.14]; LMM *z* = 0.89, *p* = 0.373; Figure 2D).

While we have now identified reductions in excitatory synaptic signaling to CA1 pyramidal neurons following both acute or chronic neutron irradiation, we also previously observed that charged particle irradiation can selectively upregulate inhibitory signaling within the hippocampus [25,26]. Therefore, we next evaluated the impact of acute neutron irradiation on the inhibitory postsynaptic signaling received by CA1 pyramidal neurons (Figure 2E–H). Here, we detected no alterations in spontaneous inhibitory postsynaptic current (sIPSC) frequency (*M*_diff_ = −0.10 Hz, 95% CI [−0.76, 0.49]; *d* = −0.11, 95% CI [−0.89, 0.73]; LMM *z* = 0.29, *p* = 0.770; Figure 2E,F) or amplitude (*M*_diff_ = −9.10 pA, 95% CI [−23.87, 1.83]; *d* = −0.51, 95% CI [−1.12, 0.32]; LMM *z* = 0.88, *p* = 0.381; Figure 2G,H) following irradiation. Additional analysis of sEPSC and sIPSC properties did not reveal any additional radiation-induced disruptions (Appendix A).

In neutron-irradiated CA1 pyramidal neurons, coincident reductions in excitatory synaptic inputs and elevated excitability in response to current injections are consistent with the action of intrinsic and synaptic homeostatic mechanisms that attempt to maintain hippocampal network stability [40,41,42]. Such homeostatic changes can be multifaceted, involving alterations to synaptic proteins [43,44,45] and voltage-gated channels [46,47,48]. While homeostatic mechanisms may be sufficient to balance out perturbations in neuronal network signaling under baseline conditions, they may still push the neutron-irradiated network into a state that is less able to deal with further challenges [49,50]. To further understand the impact of acute neutron irradiation on neuronal network function, we next assessed long-term potentiation within the hippocampus.

### 2.2. Hippocampal Long-Term Synaptic Plasticity Is Not Disrupted following Acute Neutron Irradiation

The hippocampus is comprised of an intricate network of connections between excitatory neurons and diverse populations of GABAergic interneurons that maintains balance through activity-dependent synaptic plasticity mechanisms. Long-term potentiation (LTP) of synapses from CA3 onto CA1 pyramidal neurons constitutes a critical cellular-level process for memory formation [51,52]. Due to the reduced excitatory synaptic inputs to CA1 neurons following acute neutron irradiation, we evaluated whether LTP-associated synaptic plasticity mechanisms were also impaired. 

We examined LTP induction in acute hippocampal slices prepared 3 months following irradiation. LTP was induced with theta burst stimulation (TBS) of the Schaffer collaterals and then quantified as the relative change in the slope of evoked field excitatory postsynaptic potentials (fEPSPs) generated by CA1 apical dendrites (Figure 3A). Such relatively mild stimulation produced stable LTP that manifested as a similarly elevated fEPSP slope at 60 min post-TBS in both control (150.7 ± 3.01% baseline, 95% CI [144.1, 157.4]) and acutely neutron-irradiated hippocampi (153.2 ± 2.37% baseline, 95%CI [147.8, 158.5]; *t*-test, *p* = 0.543; Figure 3B). The equivalent LTP between groups did not appear to be due to the specific stimulus intensities applied, as the slope of the relationship between fiber volley amplitude and fEPSP slope was similar between recordings from control (7.53 ± 0.79) and neutron-irradiated mice (6.07 ± 0.46; *p* = 0.125; Figure 3C). We were also unable to detect any differences in presynaptic plasticity of neurotransmitter release during paired-pulse facilitation due to acute neutron irradiation (F_(1,22)_ = 0.59, *p* = 0.452, two-way repeated measures ANOVA; Figure 3D).

Overall, we did not observe any conspicuous alterations in the network-level synaptic plasticity properties within hippocampal circuits resulting from acute neutron irradiation. Nevertheless, alterations in hippocampal mechanisms sufficient to induce memory deficits can occur even in the absence of clearly altered LTP properties [52,53,54]. Therefore, we next evaluated how acute neutron irradiation impacted overall learning and memory behavior.

### 2.3. Acute Neutron Irradiated Induces Persistent Learning and Memory Deficits

Expanding beyond the scope of our prior assays that focused specifically on the hippocampal impacts of acute neutron irradiation, behavioral tasks often require the involvement of a wider subset of brain regions. We previously determined that several cognitive processes become persistently disrupted by chronic exposures to neutron irradiation [35]. Therefore, we next conducted a battery of behavioral tasks to better understand the persistent cognitive deficits that arise following acute neutron irradiation and therefore may pose an elevated risk to astronauts during future space exploration missions.

Behavior involving recognition memory, including the novel object recognition (NOR) and object in place (OiP) tasks, is known to require proper signaling among both the hippocampus and other regions of the central nervous system [55,56]. In our initial NOR testing, the ability of neutron-irradiated animals to differentiate between familiar and novel objects (10.19 ± 3.68, 95% CI [2.164, 18.22]) was similar to the performance of control animals (11.55 ± 2.41, 95% CI [6.42, 16.68]; *p* = 0.846; Figure 4A). However, in the OiP task, mice that received acute neutron irradiation had a diminished capacity to differentiate objects that were relocated to a new location (−2.57 ± 2.49, 95% CI [−7.99, 2.84]) than were control mice (10.44 ± 2.65, 95% CI [4.82, 16.07]; *p* = 0.0028; Figure 4B). Proper NOR behavior depends heavily upon the perirhinal cortex, with lesser involvement of other brain regions, such as the hippocampus [57,58,59]. Conversely, OiP performance requires coordinated activity across several brain regions, including the medial prefrontal cortex, perirhinal cortex, hippocampus and medial thalamus [60,61,62,63]. Since OiP behavior places greater demands upon memory systems than the NOR task [62], there are more critical OiP-associated nervous system elements that can become disrupted, leading to the greater observed sensitivity to acute neutron irradiation.

We additionally performed a complementary set of experiments to measure whether associative learning and extinction of fear memories becomes altered by acute neutron irradiation. During the initial conditioning phase, when mice were trained to associate tone presentation with the delivery of a foot shock, mice in general displayed increased freezing behavior with each of three tone-shock pairings (F_(2,32)_ = 102.6, *p* < 0.001, two-way repeated measures ANOVA; Figure 4C). Initial fear acquisition was equivalent between control and neutron-irradiated animals (F_(1,29)_ = 0.630, *p* = 0.434, two-way repeated measures ANOVA). Over three subsequent days of fear extinction, where mice were presented with unpaired tones in a new context, freezing responses failed to decline as substantially in acutely neutron irradiated mice as they did in controls (F_(1, 26)_ = 16.15, *p* < 0.001, two-way repeated measures ANOVA). An interaction between treatment and day of fear extinction (F_(2, 52)_ = 4.54, *p* = 0.020, two-way repeated measures ANOVA) showed that responses in neutron irradiated mice became noticeably worse by day 2 of the extinction trials (Day 2: *p* = 0.003; Day 3: *p* = 0.009; Bonferroni *post hoc* testing). During a subsequent day of extinction testing where only 3 tones were presented, fear responses remained greatly elevated in the mice that received acute neutron irradiation (21.9 ± 3.6% of time freezing, 95% CI [14.29, 29.73]), relative to control animals (8.2 ± 1.6% of time freezing, 95% CI [4.84, 11.66]; *p* = 0.002; Figure 4D). Impaired fear extinction can result from disruptions to normal function of the hippocampus, medial prefrontal cortex and amygdala [64,65], and is also associated with post-traumatic stress disorder-like behavioral phenotypes [65,66].

The deficits in recognition and fear memory performance that arise following acute neutron irradiation are largely similar to those induced by chronic neutron exposures [35]. Thus, both irradiation paradigms appear to disrupt cognitive processes that require proper coordination of network activity across multiple brain regions. Therefore, we next investigated the impacts of acute neutron irradiation on a wider set of behavioral characteristics.

### 2.4. Social and Internalizing Behaviors Are Not Altered by Acute Neutron Irradiation

In our final set of experiments, we assessed how acute neutron irradiation altered behaviors associated with social interactions and internalizing disorders. We previously observed that both anxiety-like and social avoidance behaviors become elevated following chronic neutron irradiation [35].

To assess social interaction behavior, mice that were previously habituated with cage mates were placed in a barrier-free arena and allowed to freely interact with a novel mouse. Neutron-irradiated mice spent a similar overall amount of time engaged in social interactions with the novel mouse (12.10 ± 0.58 s, 95% CI [10.84, 13.36]) as did control animals (12.12 ± 0.43 s, 95% CI [11.21, 13.04]; *p* = 0.953; Figure 5A). We also did not observe any difference in the time spent actively avoiding interactions with the novel animal by either the acutely neutron-irradiated (3.58 ± 0.23 s, 95% CI [3.08, 4.07]) or control mice (3.48 ± 0.18 s, 95% CI [3.11, 3.85]; *p* = 0.891; Figure 5B). Social interaction behaviors are known to involve diverse neuronal networks, including the medial prefrontal cortex, hippocampus and ventral tegmental area [55,56,67].

Lastly, we evaluated the impact of acute neutron irradiation on the internalizing behavior of animals, testing for changes in anxiety- and depression-like behaviors. Acute exposures to low doses of charged particle irradiation are sufficient to produce persistent increases in both anxiety- and depression-like behavior in mice [9,68]. The light-dark box assay measures elevated anxiety in animals through their increased avoidance of the more brightly lit compartment of the testing arena [69]. However, mice that received acute neutron irradiation performed a similar number of transitions between light and dark compartments (15.00 ± 1.12, 95% CI [12.57, 17.43]) as control animals (13.76 ± 1.01, 95% CI [11.63, 15.90]; *p* = 0.414; Figure 5C), suggesting that anxiety-like behavior was unaltered. The relative time mice spent in each of the chambers was similarly unchanged by acute neutron irradiation (*p =* 0.799; data not shown). We also did not observe any depression-like behavior during forced swim testing, with control animals spending an equivalent amount of the total trial time immobile (44.08 ± 2.72 s, 95% CI [38.32, 49.84]) as the mice that received acute neutron irradiation (46.88 ± 2.97 s, 95% CI [40.47, 53.29]; *p* = 0.262; Figure 5D).

## 3. Discussion

The goal of the current study was to develop a better understanding of the neurological risks associated with low-dose neutron irradiation, particularly in the context of our prior findings [35] that chronic neutron exposures persistently induce both neurobehavioral and electrophysiological defects in mice. Utilizing electrophysiological recordings, we identified similar cellular-level alterations in hippocampal neuron function following acute neutron irradiation, yet more modest changes in network-level signaling. Through behavioral testing with a similar battery of assays, we observed multiple neurological deficits after acute neutron irradiation, although to a more limited extent than what occurred following chronic neutron exposures. Overall, we find that although the risks are perhaps not as grave as following chronic neutron exposures, equivalent acute neutron irradiation doses remain a substantial hazard to diverse aspects of nervous system function.

For unfortunate reasons, there is already some understanding of the serious potential human health risks associated with acute neutron radiation exposures, gleaned from studies that tracked atomic bomb survivors. Survivors from Hiroshima and Nagasaki, where a meaningful portion of the total radiation dose came from neutrons, have displayed long-term elevated risks for developing cancer, cardiovascular disease and neurodevelopmental disruptions [70,71,72]. Focusing on risks to the central nervous system, animal experiments indicate that mixed-fields of γ-ray and neutron irradiation, as occurred acutely in atomic bomb exposures, reduce hippocampal neurogenesis [73]. When received at similar doses, neutron radiation is substantially more effective than γ-rays at suppressing adult neurogenesis and triggering neuronal apoptosis [74,75,76]. However, prior studies did not explore how anatomical changes in the brain following acute neutron irradiation may also correspond with alterations in the functional activity of neurons and neural networks.

Therefore, instead of focusing on the relatively small number of neurons that undergo apoptosis following space-relevant doses of acute neutron irradiation, we decided to undertake the first assessment of how the signaling properties among the much larger population of surviving neurons become altered. Our data indicate that acute neutron irradiation increases the rate at which hippocampal neurons generate action potentials in response to excitatory inputs. Similar increases in hippocampal excitability are associated with disorders such as epilepsy [77] and intellectual disability [78]. Alterations in voltage-gated ion channels expression, such as occurs after similar doses of charged particle radiation [24,79], could underlie the elevated neuronal responses. Radiation exposures can alter ion channel expression though direct interactions [80], generation of reactive oxygen species [6,24], alteration of epigenetic regulation [81] or due to homeostatic plasticity mechanisms that work to counteract changes in synaptic inputs [46,47,48]. Although the other intrinsic properties we surveyed following acute neutron irradiation appeared to be largely unchanged, regulation of neuronal excitability is highly multifaceted and neurons are capable of maintaining similar outward properties despite numerous changes in underlying molecular mechanisms [50]. Therefore, we believe that changes in synaptic signaling properties may more fully reflect the impacts of acute neutron radiation on hippocampal network function.

While the specific consequences of acute neutron irradiation on neuronal morphology remains unexplored, chronic neutron exposures reduce hippocampal synaptic densities, including those of more mature, mushroom-like spines [82]. Such changes are consistent with how dendritic complexity and spine density both decrease following acute, low-dose, exposures to single- [12,14,36,68,83] and dual-ion GCR irradiations [20,22]. Dendritic spines are known to predominantly contain excitatory synapses [84,85], so our results showing preferential decreases in sEPSC frequency are consistent with other forms of particle radiation causing damage to dendritic spines. Although proton and multi-ion GCR irradiation can elevate GABAergic signaling from hippocampal interneurons [25,26], we do not see clear evidence that such is the case following acute neutron irradiation. Furthermore, reduced sEPSC frequency is similar to what we observe following both chronic neutron [35] and acute ^4^He irradiation [9]. Such disruptions to excitatory synaptic signaling pathways may critically alter the generation of rhythmic oscillations within the hippocampus and could therefore lead to perturbation of associated memory functions [59,86].

Given the deficits we measured in CA1 synaptic signaling following acute neutron irradiation, we were surprised to not observe any corresponding alterations in hippocampal LTP capabilities. Indeed, we had previously determined that changes to hippocampal synaptic inputs were accompanied by LTP deficits following either acute multi-ion GCR or chronic neutron exposures [26,35,87]. One key mechanism of LTP is through regulation of postsynaptic glutamate receptors [88,89,90]. However, there appears to be substantial variability in how different acute GCR exposure paradigms alter expression of specific excitatory synaptic signaling components. Downregulation of GluA1 and GluN1 receptors following 10 cGy ^16^O irradiation [12], GluN2B after combined 60 cGy proton + ^16^O irradiation [22], or GluA1 synaptic surface expression subsequent to 50 cGy 3-ion GCR exposure [21] would all be consistent with weakened LTP. Yet, increased GluA1 phosphorylation after 50 cGy proton irradiation [6], and elevated GluN1, GluN2A and GluN2B expression following 5 cGy ^16^O irradiation [14] would both seem likely to facilitate LTP induction. Therefore, the net impact of various acute neutron irradiation-induced disruptions of glutamatergic signaling mechanisms may occlude any overall change in hippocampal LTP. Another possible reason that reductions in excitatory synaptic inputs to CA1 were not accompanied by measurable LTP changes is that while acute neutron radiation may have eliminated a subset of glutamatergic connections, the remaining synapses may have retained normal plasticity responses.

Although we did not observe altered hippocampal LTP following acute neutron irradiation, animals did appear to display substantial deficits in cognitive performance. As we mentioned above, our LTP results only indicate that the synaptic connections that remain after acute neutron irradiation retain normal plasticity responses, but they did not assess how a net reduction in excitatory synaptic inputs may have undermined the neuronal circuits required for learning and memory processes. Indeed, manipulations that lead to reduced dendritic spine densities, such as deletion of *Stim* Ca^2+^ sensors genes, can result in memory impairments, while LTP induction conversely becomes simultaneously enhanced [91]. Additionally, while our electrophysiological studies focused on hippocampal changes, many of our behavioral assays depend on the proper activity across multiple brain regions, which may show varied responses to acute neutron irradiation. Whereas NOR behavior depends mostly upon proper hippocampal and perirhinal cortex function, additive or more severe acute neutron radiation-induced disruptions in medial prefrontal cortex and medial thalamus structures required for OiP performance could explain why the latter task displays a greater detriment [62]. Likewise, fear extinction behavior may be more vulnerable to diffuse radiation injuries, as hippocampus, amygdala and medial prefrontal cortex involvement are all required for normal behavior [64,65]. Our findings therefore suggest that complex behaviors involving increased task rigor are more susceptible to radiation-induced changes, and represent the types of tasks likely to be critical for astronauts on future exploratory missions.

One key question our current study considers is the relative impact of radiation dose-rate on the occurrence of subsequent neurological disruptions. Our prior study examining the neurocognitive impacts of chronic, low dose-rate, neutron irradiation was the first to explore how multiple levels of brain function become perturbed even when radiation exposures extend over space mission-relevant time courses [35]. To contextualize those results, we applied a similar neutron irradiation paradigm that mainly differed in delivering the entire radiation dose within an acute time frame that more closely matches the overwhelming majority of other GCR studies. We were interested to see that low-dose neutron irradiation shows many time course-invariant similarities in how the central nervous system is impacted. Chronic and acute neutron irradiation each reduced the frequency of excitatory synaptic signaling received by hippocampal neurons, and altered output excitability. Both neutron dose rates also induced similar deficits in spatial and fear extinction memory functions. The main differences appeared in how chronic neutron irradiation was more likely to disturb LTP-related network plasticity and social interaction behaviors. Thus, while more acute exposure paradigms appear sufficient to study the broad hazards of space-relevant radiation responses, chronic irradiation studies seem necessary to fully understand the risks that astronauts will face.

While neutrons represent only one element of the overall space radiation environment, shielding interactions will further elevate their proportion within the spacecraft effective radiation field [2,27,28] and neutrons remain one of the best proxies for modeling chronic particle radiation exposures [34]. Our study provides an important opportunity to compare the impacts of acute neutron irradiation against prior central nervous system disruptions observed after acute charged particle GCR exposure models. While acute neutron irradiation produces similar decreases in excitatory synaptic signaling frequency to acute ^4^He irradiation [9], we have instead observed preferential increases in inhibitory synaptic signaling in multiple other acute charged-particle irradiation models [25,26]. Thus, while neutron exposures pose a broadly similar risk to alter synaptic connectivity, further research is needed to better understand the mechanisms through which specific GCR particle combinations differentially alter neuronal structures, channels and signaling pathways. Finally, there is now extensive evidence that various space-relevant charged particle GCR exposures induce behavioral alterations in animals see review: [92]. What has become clear is that even when a particular irradiation model is replicated, the exact same behavioral changes are not always observable, and do not always translate across a given cohort (i.e., the same animals are not uniformly disrupted across all behavioral tasks administered). For example, low dose (10–50 cGy) proton irradiation is sufficient to cause spatial memory deficits [6,8], yet the same exposures have no impact [93], or even enhance [8] contextual fear memory. Despite such heterogeneity in behavioral responses to irradiation, acute neutron irradiation appears to induce a similar spectrum of behavior deficits that commonly arise following acute charged particle radiation exposures. We therefore believe that our finding that acute, low-dose, neutron irradiation negatively alters several levels of central nervous system function is important for both understanding the risks posed by an important element of the spacecraft radiation field and provides useful insight into the general hazards of GCR exposures. Continuing to develop a more comprehensive understanding of the manner in which different space radiation components alter cognitive function is critical for properly assessing the risks astronauts will face on future space missions and for enabling the development of appropriate protective strategies.

## 4. Materials and Methods

### 4.1. Animals

Male C57BL/6J mice (JAX) were utilized in these studies and were 6 months old at the time of irradiation. All experiments were approved by the Institutional Care and Use Committees at Columbia University, Stanford University (Protocol 30183, approved 7 July 2021) and the University of California, Irvine (Protocol 21-025, approved 16 March 2021; 20-095, approved 4 September 2020). Procedures involving animals all conform to National Institute of Health and institutional guidelines. Mice were group housed (2–4 per cage), received *ad libitum* access to food and water, and were maintained on a 12 h light/dark cycle throughout the study. A total of 53 mice, split across 3 cohorts, were used in the experiments described below (Figure 6). 

### 4.2. Neutron Irradiation and Dosimetry

Irradiation was performed in the neutron irradiator at the Columbia University Radiological Research Accelerator Facility (RARAF; Irvington, NY, USA) using established approaches [94,95]. Briefly, neutrons were generated by impinging a mixed beam of 5 MeV protons, deuterons and molecular ions on a thick beryllium target [96], with the mice positioned 190 mm away from the target at an angle of 60° to the primary beam. Mice were restrained within 37 mm × 37 mm × 68 mm polystyrene enclosures. The enclosures were then suspended from a Ferris wheel apparatus, allowing the mice to be slowly rotated at a rate of 0.5 rotations per min around the neutron beam, providing isotropic irradiation. A total radiation dose of 18 cGy was delivered at a dose-rate of 0.5 Gy/h. Halfway through delivery of the total dose, the enclosures were rotated front-to-back, to further ensure the uniformity of irradiation. While the delivered radiation dose was composed primarily of neutrons with a broad spectrum of energies (0.2–9 MeV), ~19% consisted of inherent γ-rays [97].

Total dose was measured using a custom tissue-equivalent (TE) gas ionization chamber [98], filled with ~700 Torr methane-based TE gas and calibrated prior to each use, using a NIST-traceable 50 mg ^226^Ra gamma-ray source. Gamma-ray dosimetry was performed separately with a custom-made compensated Geiger-Mueller dosimeter, which has very low sensitivity to neutrons. Both detectors were sequentially placed on the Ferris wheel and rotated around the neutron beam, mimicking the mouse irradiations. The measured dose was used to calibrate a second TE gas ionization chamber, which was permanently located at an angle of 15° to the primary beam axis and used as a beam monitor, to terminate irradiations when the prescribed dose was achieved.

No apparent radiotoxicity was observed in mice following irradiation. Age-matched control mice underwent all aspects of the study in parallel to those receiving irradiation, were housed under identical conditions and handled equivalently, aside from not receiving neutron irradiation.

### 4.3. Whole Cell Electrophysiology

Between 3–5 months following the conclusion of irradiation, male mice were deeply anesthetized by Ketamine/Xylazine, Patterson Veterinary Supply Inc. Devens, MA, USA) and then transcardially perfused with an ice-cold protective recovery solution containing (in mM): 92 NMDG, 26 NaHCO_3_, 25 glucose, 20 HEPES, 10 MgSO_4_, 5 Na-ascorbate, 3 Na-pyruvate, 2.5 KCl, 2 thiourea, 1.25 NaH_2_PO_4_, 0.5 CaCl_2_, titrated to a pH of 7.3–7.4 with HCl (all chemical reagents are from Sigma-Aldrich, St. Louis, MO, USA, unless otherwise noted) [99]. Coronal slices (300 µm) containing the hippocampus were cut in ice-cold protective recovery solution using a vibratome (VT1200S, Leica Biosystems, Buffalo Grove, IL, USA). Brain slices were then incubated in 35 °C protective recovery solution for 12 min. Subsequently, brain slices were maintained in room temperature artificial cerebrospinal fluid (aCSF) consisting of (in mM): 126 NaCl, 26 NaHCO_3_, 10 glucose, 2.5 KCl, 2 MgCl_2_, 2 CaCl_2_, 1.25 NaH_2_PO_4_. All solutions were equilibrated with 95% O_2_/5% CO_2_.

Intracellular recordings were performed in a submerged chamber perfused with oxygenated aCSF at 2.5 mL/min and maintained at 33 °C by a chamber heater (BadController V, Luigs and Neumann, Ratingen, Germany). Hippocampal neurons were visualized using DIC illumination on an Olympus BX61WI microscope (Olympus Microscopy, Waltham, MA, USA) with an sCMOS camera (Flash 4.0 LT+, Hamamatsu, Bridewater, NJ, USA). Recording pipettes were pulled from thin-walled borosilicate capillary glass (King Precision Glass, Claremont, CA, USA) using a P97 puller (Sutter Instruments, Novato, CA, USA) and were filled with (in mM): 126 K-gluconate, 10 HEPES, 4 KCl, 4 ATP-Mg, 0.3 GTP-Na, 10 phosphocreatine (pH-adjusted to 7.3 with KOH, osmolarity 290 mOsm). Pipettes had a 2.5–5 MΩ tip resistance.

Whole cell recordings were performed on CA1 superficial layer pyramidal neurons in the dorsal hippocampus. Firing properties were assessed during current injection steps (−100 to 350 pA, 1s). Recordings were excluded for neurons with a resting membrane potential above −55 mV or where the series resistance increased by >20% of baseline. Pipette capacitance was neutralized for all recordings. Input resistance was calculated from the change in steady-state membrane potential resulting from hyperpolarizing current injections, while sag was measured as the difference between the steady-state and peak negative potential during a −100 pA hyperpolarizing current injection. Action potential threshold was the voltage where the *dV*/*dt* prior to a detected event first exceeded 3 times the standard deviation. Width was the time an action potential, resampled at 100 kHz, exceeded the half-height between threshold and peak voltages, and cells with a width < 0.75 ms were excluded as potential fast-spiking interneurons. Action potential properties were only measured in the first spike evoked by a depolarizing current for each neuron. Spontaneous excitatory postsynaptic current (sEPSC) activity was measured as inward currents while neurons were held at −65 mV, whereas spontaneous excitatory postsynaptic currents (sIPSCs) were outward currents observed in neurons held at 0 mV. Charge transfer was calculated by integrating the area of postsynaptic currents and rise time was the time required to increase from 10% to 90% of peak amplitude. Events with a rise time > 7.5 ms, a peak amplitude of <3 pA or a charge transfer of <25 pC were excluded.

Data were acquired in pClamp software (Molecular Devices, San Jose, CA, USA) using a Multiclamp 700B amplifier (Molecular Devices), low-pass filtered at 2 kHz, and digitized at 10 kHz (Digidata 1440A, Molecular Devices). Data analysis was performed using Clampfit (Molecular Devices) or custom written Python (Fredericksburg, VA, USA) scripts. *n* = 5 control mice yielded recordings from *n* = 16/15 cells (for intrinsic and synaptic properties, respectively). Likewise, *n* = 5 neutron-irradiated mice yielded recordings from *n* = 14/14 cells (for intrinsic and synaptic properties, respectively).

### 4.4. Extracellular Field Recordings

Hippocampal slices were prepared as previously described [100] at 3 months following irradiation. Following isoflurane anesthesia, mice were decapitated and the brain was quickly removed and submerged in ice-cold, oxygenated dissection medium containing (in mM): 124 NaCl, 3 KCl, 1.25 KH_2_PO_4_, 5 MgSO_4_, 26 NaHCO_3_, and 10 glucose. Coronal hippocampal slices (320 µm) were prepared using a vibratome (VT1000S, Leica Biosystems) before being transferred to an interface recording chamber containing preheated aCSF consisting of (in mM): 124 NaCl, 3 KCl, 1.25 KH_2_PO_4_, 1.5 MgSO_4_, 2.5 CaCl_2_, 26 NaHCO_3_, and 10 glucose and maintained at 31 ± 1 °C. Slices were continuously perfused with this solution at a rate of 1.75–2 mL/min while the surface of the slices were exposed to warm, humidified 95% O_2/_5% CO_2_. Recordings began following at least 2 h of incubation.

Field excitatory postsynaptic potentials (fEPSPs) were recorded from CA1b stratum radiatum apical dendrites using a 2M NaCl-filled glass pipette (World Precision Instruments, Sarasota, FL, USA) (2–3 MΩ) in response to orthodromic stimulation (twisted nichrome wire, 65 µm diameter) of Schaffer collateral-commissural projections in CA1 stratum radiatum. Input/output curves were initially generated using 10 µA steps and a current that elicited a 50% maximal spike-free response was used for all subsequent stimulations in that slice. Paired-pulse facilitation was next assessed as the relative change in fEPSP slope when stimuli were delivered at 40, 100 and 200 ms intervals. With pulses administered at 0.05 Hz, a 20 min stable baseline was established and then long-term potentiation (LTP) was induced by theta burst stimulation (TBS), consisting of 5 bursts containing four pulses at 100 Hz, with a 200 ms inter-burst interval. The stimulation intensity was not increased during TBS. Data were collected and digitized by a NAC 2.0 Neurodata Acquisition System (Theta Burst Corp., Irvine, CA, USA) and stored on a disk.

Data in the text are presented as means ± SD, while in the figures as mean ± SEM. The fEPSP slope was measured at 10–90% fall of the slope and data in figures on LTP were normalized to the last 10 min of baseline. Electrophysiological measures were analyzed using a Student’s *t*-test, *p* < 0.05. *n* = 12/10 hippocampal slices from *n* = 6/6 animals (control and irradiated, respectively).

### 4.5. Behavioral Testing

Concurrent behavioral testing of control (*n* = 17) and irradiated mice (*n* = 14) occurred across four weeks, beginning two months after acute neutron irradiation. An investigator blinded to the experimental cohorts performed all behavioral testing.

#### 4.5.1. Episodic and Spatial Memory Testing

Novel object recognition (NOR) and object in place (OiP) spontaneous exploration tasks rely on intact hippocampal, medial prefrontal cortex and perirhinal cortex function [60,101]. The NOR task evaluates episodic recognition memory through measuring the preference of mice to investigate novel environmental changes, whereas the OiP task evaluates associative recognition memory. Both tasks were conducted between 11–13 weeks after irradiation, as described previously [6]. Briefly, NOR and OiP testing occurred in a dimly lit (48 lux) test arena (30 cm × 30 cm × 30 cm) with a layer of fresh bedding that was filmed from above. All bedding was replaced and the arena was thoroughly cleaned with 70% ethanol between trials. For the NOR task, mice were initially habituated to the empty arena for three days (10 min/day). The following testing day, two plastic objects (differing in color, shape and size) were magnetically affixed 16 cm apart in the arena and the mouse was allowed five minutes to explore the objects. The mouse was returned to the home cage for five minutes while one familiar object was substituted for a novel object (both objects were cleansed with 70% ethanol). The mouse was then returned to the arena for five minutes of further exploration. OiP testing began one week after the NOR task, with two days (10 min/day) of habituation. On the third day, mice explored an arena with four unique objects for five minutes before briefly returning to their home cage (5 min). All objects were cleansed with 70% ethanol and the location of two objects was swapped before the mouse was returned for five more minutes of exploration. Video of both tasks was scored for time spent interacting (nose within 2 cm) with familiar versus novel (or relocated) objects. The discrimination index was then calculated for each mouse from these values:
[(novel/total exploration time) − (familiar/total exploration time)] × 100


#### 4.5.2. Social Interaction Testing

Social interaction and social avoidance behaviors were evaluated in mice 13 weeks after irradiation, using established protocols [67,102,103]. Mice were initially each individually habituated to the well-lit (915 lux) test arena (30 cm × 30 cm) for two days (15 min/day). On the third day of the trial, the test mouse was allowed to explore freely for 10 min, prior to a novel mouse (3-month old, C57BL6/J male, weighing less than the test mouse) being introduced into the arena. The mice were allowed to explore and interact freely without barrier for 10 min, and active interaction or avoidance was recorded. Social interactions included any time the test mouse spent sniffing while in active contact with the novel animal’s snout, flank, or anogenital area, mutual grooming, or directed pursuit of the novel mouse. Concurrently, avoidance behavior was characterized as time the test mouse spent actively avoiding social interactions initiated by the novel mouse.

#### 4.5.3. Anxiety- and Depression-Like Behavior Testing

After completion of NOR and OiP testing, mice were evaluated 13 weeks after irradiation for anxiety-like behavior with the light-dark box (LDB) test and depressive-like behavior with the forced swim test (FST), using established methods [9,69,104]. The LDB arena consisted of a light compartment (30 cm × 20 cm × 27 cm, 915 lux) connected to a dark compartment (15 cm × 10 cm × 27 cm, 4 lux) via a small opening (7.5 cm × 7.5 cm). Thus, the LDB test contrasts the natural propensity of mice to explore new environments with their degree of anxiety to be in a well-lit space. Mice were placed in the arena for 10 min, and we recorded time spent in each chamber and number of transitions between compartments. In the FST task, we evaluated hippocampal-dependent, depression-like, behavior by measuring the responses of mice to being placed in a tank of water (15 cm diameter × 20 cm, 24 °C) for five minutes. We quantified the amount of time each mouse spent floating (despair-like behavior) versus swimming or climbing during the entire 5 min duration.

#### 4.5.4. Fear Extinction Testing

To test whether mice could learn and later extinguish conditioned fear responses, we performed a series of established fear extinction (**FE**) assays at 15 weeks after irradiation, as adapted [35,65,105]. Testing occurred in two similar contexts within a behavioral conditioning chamber (17.5 cm × 17.5 cm × 18 cm, Coulbourn Instruments, Allentown, PA, USA) with a steel slat floors (3.2 mm diameter slats, 8 mm spacing). For context A the chamber was scented with a spray of 10% acetic acid in water, while in context B the steel floor of the chamber was covered with white plastic and a spray of 10% almond extract in water was applied. Initial fear conditioning was performed in context A after mice were allowed to habituate to the chamber for two minutes. Three pairings (spaced by 120 s) of an auditory conditioned stimulus (16 kHz tone, 80 dB, lasting 120 s; CS), co-terminating with a foot-shock unconditioned stimulus (0.6 mA, 1 s; US) were presented. On the following three days of extinction training, mice were initially habituated to context B for two minutes before being presented with 20 non-US reinforced CS tones (16 kHz, 80 dB, lasting 120 s, at 5 s intervals). On a final day of fear testing mice were presented with only three non-US reinforced CS tones (16 kHz, 80 dB, lasting 120 s) at a two-minute intertrial interval in context B. Freezing behavior was recorded with a camera mounted above the chamber and scored by an automated, video-based, motion detection program (FreezeFrame, Coulbourn Instruments). FreezeFrame algorithms calculate a motion index for each frame of the video, with higher values representing greater motion. An investigator blinded to the experimental groups set the motion index threshold representing immobility for each animal individually, based on identifying a trough separating low values during immobility and higher values associated with motion. Freezing behavior was defined as continuous bouts of 1 s or more of immobility. The percentage of time each mouse spent freezing was then calculated for each phase of the fear response testing.

## 5. Statistical Analyses

The level of significance for behavioral testing was assessed by Mann-Whitney’s two-tailed, non-parametric *t* test using Prism data analysis software (v8, GraphPad, San Diego, CA, USA). For the fear extinction experiment, conditioning day 1 (T_1_–T_3_) and extinction training days 1–3 were analyzed using two-way ANOVAs followed by Bonferroni’s *post hoc* tests with freezing time (%) as within-subjects variables and group/irradiation treatment (0 cGy controls vs. 18 cGy). For behavioral testing, animals with measurements >2 SD beyond the group mean in a given assay were excluded from the final analysis. To account for the nested data produced by whole cell electrophysiology recordings, differences between treatment groups were evaluated by a linear mixed-effect model regression analysis [37] run in Python. Calculation of estimation statistics-based confidence intervals was performed with the DABEST package in python [38]. Gardner-Altman estimation plots include a 5000 resampling, bias-corrected and accelerated bootstrap analysis to determine the nonparametric confidence interval of differences between groups. We quantified effect sizes with an unbiased *Cohen’s d* test. Results were expressed as mean values ± SEM, unless stated otherwise, and *p* values of ≤0.05 were considered statistically significant.

## Figures and Tables

**Figure 1 ijms-22-09020-f001:**
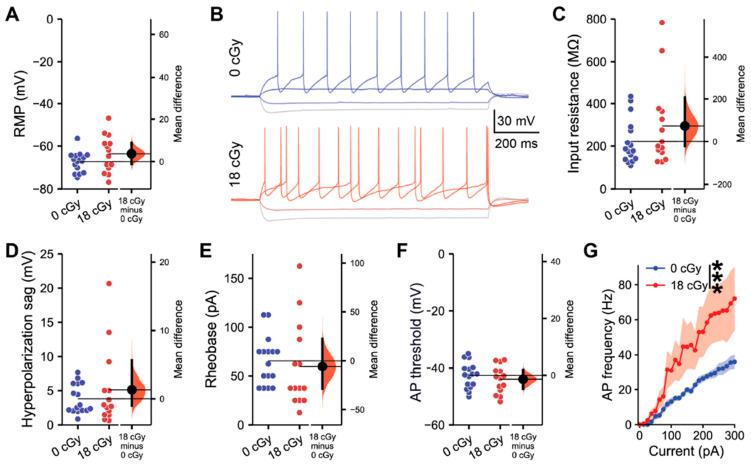
Acute, low-dose, neutron irradiation increases the intrinsic excitability of CA1 pyramidal neurons. All data are from whole cell current clamp recordings of CA1 pyramidal neurons from the superficial layer of the dorsal hippocampus, 3–5 months after acute exposure to 18 cGy neutron irradiation. (**A**) Resting membrane potential (RMP) was unchanged between groups. (**B**) Representative examples of responses to a range of brief current injections in 0 cGy and 18 cGy neurons. There was no alteration in either the input resistance (**C**) or sag during a −100 pA hyperpolarizing current injection (**D**) between treatment groups. Both the rheobase current (**E**) and the threshold potential (**F**) required for action potential (AP) initiation also remained unchanged. (**G**) Across a range of current injections, neutron irradiated neurons generated more frequent APs. *n* = 5/5 animals, 16/14 cells (0 cGy and 18 cGy, respectively) for grouped data. Gardner-Altman estimation plots show raw data on the left axis and a bootstrapped sampling distribution on the right axis. A black dot depicts the mean difference between groups and the 95% confidence interval is indicated by the ends of the vertical black bars. Data are presented as Mean ± SEM for (**G**). *** *p* < 0.001 (two-way ANOVA).

**Figure 2 ijms-22-09020-f002:**
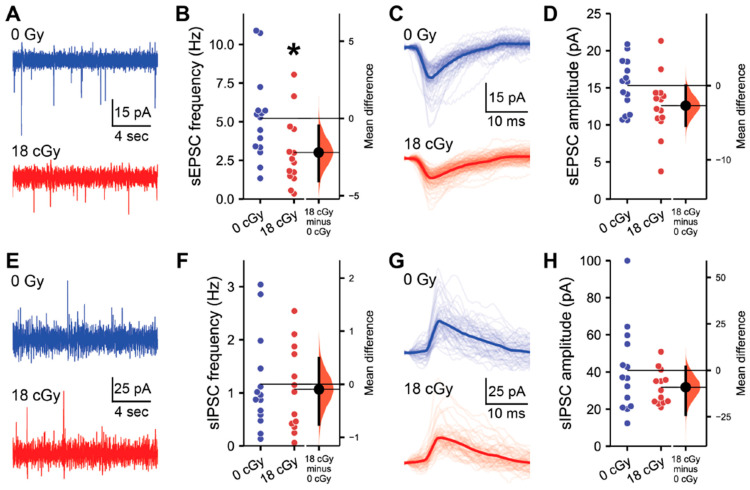
Hippocampal excitatory synaptic signaling is suppressed by acute neutron irradiation. All data are from whole cell voltage clamp recordings of CA1 pyramidal neurons from the superficial layer of the dorsal hippocampus, 3–5 months after acute exposure to 18 cGy neutron irradiation. (**A**) Representative examples of spontaneous excitatory postsynaptic current (sEPSC) recordings from 0 cGy and 18 cGy neurons. (**B**) The frequency of sEPSCs was reduced in 18 cGy neurons. (**C**) Aligned examples of sEPSCs in representative 0 Gy and 18 cGy neurons. Light lines show individual sEPSCs, while the darker line displays the average sEPSC during a 200 s recording from that neuron. (**D**) sEPSC amplitude was also similar between groups. (**E**) Representative examples of spontaneous inhibitory postsynaptic current (sIPSC) recordings from 0 cGy and 18 cGy neurons. (**F**) sIPSC frequency was equivalent between 0 Gy and 18 cGy neurons. (**G**,**H**) sIPSC amplitude was also unchanged after irradiation. *N* = 5/5 animals, 15/14 cells for sEPSCs and *N* = 5/5 animals, 14/13 cells for sIPSCs (0 cGy and 30 cGy, respectively). Gardner-Altman estimation plots show raw data on the left axis and a bootstrapped sampling distribution on the right axis. A black dot depicts the mean difference between groups and the 95% confidence interval is indicated by the ends of the vertical black bars. * *p* < 0.05 (linear mixed-effect model regression).

**Figure 3 ijms-22-09020-f003:**
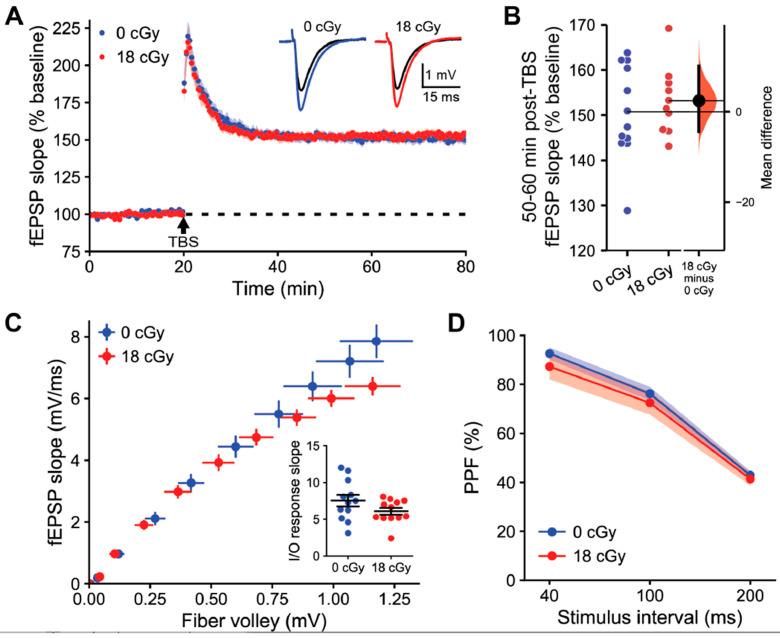
Acute neutron irradiation does not alter long-term synaptic plasticity in the hippocampal area CA1. Extracellular field recordings following stimulation of the Schaffer-commissural projections to the proximal apical dendrites of dorsal hippocampus field CA1b 3 months following 18 cGy acute neutron irradiation. (**A**) Following a stable 20 min baseline recording, a single train of theta burst stimulation (TBS) was applied and baseline recordings were continued for an additional 60 min. The time course shows that there was no difference in TBS-induced long-term potentiation (LTP) of field excitatory postsynaptic potential (fEPSP) slopes in neutron irradiated brain slices. Inset, Representative traces collected during baseline (black line) and 60 min post-TBS (colored line). (**B**) Relative shift in fEPSP slope 50–60 min post-TBS was equivalent between groups. (**C**) The relationships between stimulation intensity and fEPSP slope were not detectably different between groups. Inset, Slope of the input/output (I/O) relationship between fiber volley amplitude and fEPSP slope for each sample did not vary between groups. (**D**) Transmitter release kinetics, as assessed with paired pulse facilitation (PPF) measurements, were also comparable between treatments. Data in (**A**,**B**,**D**) are presented as Mean ± SEM. Gardner-Altman estimation plot in (**C**) shows raw data on the left axis and a bootstrapped sampling distribution on the right axis. A black dot depicts the mean difference between groups and the 95% confidence interval is indicated by the ends of the vertical black bars. *n* = 6/6 animals, 12/10 sections (0 cGy and 18 cGy, respectively).

**Figure 4 ijms-22-09020-f004:**
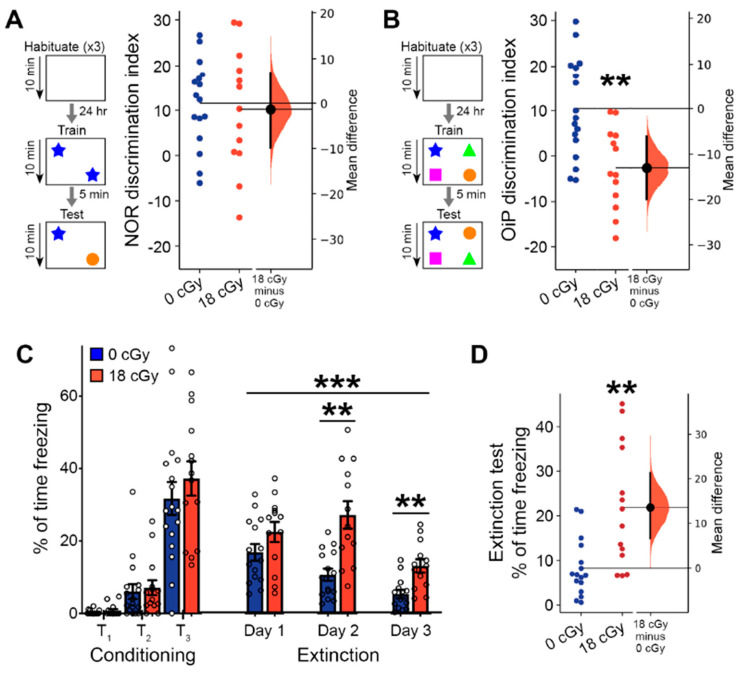
Spatial and fear extinction memory behavior is perturbed by acute neutron irradiation. Behavioral testing of memory function was performed 2–3 months after completion of the 18 cGy acute neutron irradiation. (**A**) Ability to discriminate between objects in the novel object recognition (NOR) assay was not altered in 18 cGy animals. (**B**) Object in place (OiP) differentiation of objects relocated to alternative location was diminished following acute neutron irradiation. (**C**) In a fear extinction assay, initial presentation of 3 tone-shock pairings (T_1_–T_3_) lead to increased conditioned freezing behavior. Subsequent tone-only presentations led to greater extinction of conditioned fear in control animals that diminished freezing behavior across 3 days of measurements. (**D**) After three previous days of fear extinction training, acutely neutron irradiated mice continued to show a diminished ability to extinguish past fear responses. (**A**,**B**,**D**) are Gardner-Altman estimation plots showing raw data on the left axis and a bootstrapped sampling distribution on the right axis. Black dots depict the mean difference between groups and the 95% confidence intervals are indicated by the ends of the vertical black bars. (**C**) is presented as Mean ± SEM. *n* = 17/13 animals for NOR and OiP, *n* = 16/14 animals for fear extinction (0 cGy and 18 cGy, respectively). ** *p* < 0.01, *** *p* < 0.001 (Mann-Whitney two-tailed *t*-test or two-way ANOVA).

**Figure 5 ijms-22-09020-f005:**
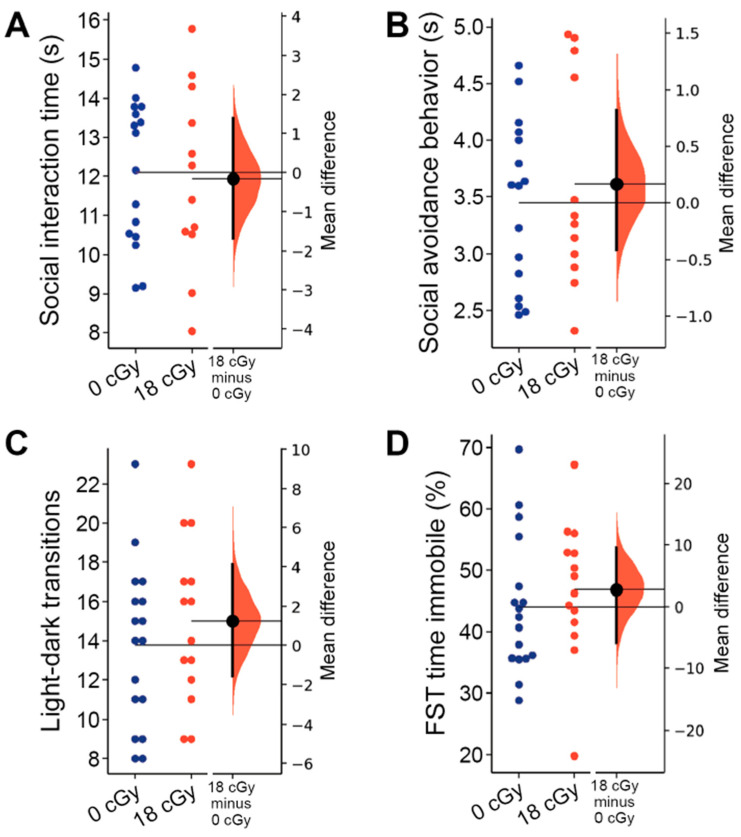
Acute neutron irradiation does not alter social or internalizing behaviors. Testing for alterations in social and internalizing behaviors was performed 2–3 months following mice received an 18 cGy acute neutron irradiation. Neutron irradiation altered neither the total time animals were involved in social interactions (**A**) nor the time experimental mice spent displaying social avoidance behaviors (**B**) during social interaction testing (SIT). (**C**) Mice in both groups performed a similar number of transitions between arena sections during light-dark box (LDB) testing. (**D**) Total time spent immobile in the forced swim test (FST) was not altered in 18cGy animals. Gardner-Altman estimation plots show raw data on the left axis and a bootstrapped sampling distribution on the right axis. A black dot depicts the mean difference between groups and the 95% confidence interval is indicated by the ends of the vertical black bars. *n* = 17/14 animals for LDB and FST, *n* = 16/12 animals for SIT (0 cGy and 30 cGy, respectively).

**Figure 6 ijms-22-09020-f006:**
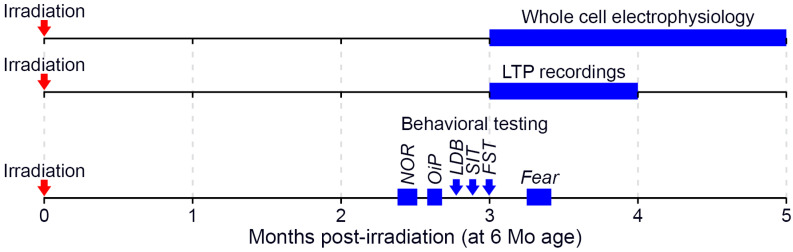
Timeline of post-irradiation experiments. Following acute neutron irradiation when 6 months old, mice were subsequently used in a variety of testing paradigms. Each horizontal timeline indicates experiments conducted in a unique and non-overlapping set of animals (LTP: long-term potentiation; NOR: novel object recognition; OiP: object in place; LDB: light-dark box; SIT: social interaction test; FST: forced swim test; Fear: fear extinction testing).

## Data Availability

Data will be made available on Zenodo, accessible on 1 December 2021.

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
