# Peer review of "Acute, Low-Dose Neutron Exposures Adversely Impact Central Nervous System Function"

_ijms, 2021, doi:10.3390/ijms22169020_

Round 1
Reviewer 1 Report
In this study, the authors have examined the effects of acute neutron exposures on mouse central nervous system parameters which complements their prior study with chronic neutron exposures. The latter study was motivated primarily by the need to examine high LET radiation effects at more space-like exposure rates. While some biological responses were similar between the two studies there were differences especially with respect to long term potentiation and social interaction behavior. And, neither neutron exposure scenario produced the same magnitude of responses observed with accelerated charged particles. So, while the neutron models recapitulate some aspects of high energy charged particle exposures they are not qualitatively identical and this is likely due to the differences in spatial patterns of energy deposition.
Neutrons primarily generate recoil protons of low energy with LET distributions peaking around 70 keV/µm. The proton tracks are very short (a few microns), produce few secondary electrons (delta rays) of significant range and do not simultaneously impact the long cylindrical groups of cells traversed by high energy accelerated ions. Both neutron generating systems (RARAF accelerator) and Colorado State University 252Cf facility produce neutrons accompanied by a significant dose contribution from gamma rays which may ~20%. A few remarks about the differences in radiation types would be helpful to the reader and the fact that the uniformly distributed neutron energy deposition events are able to produce significant CNS responses is important in the sense that spatially correlated cell group damage may be only a minor contributor to the overall high LET effects.
Technical issues
Neutrons primarily generate recoil protons of low energy with LET distributions peaking around 70 keV/µm. The proton tracks are very short (a few microns), produce few secondary electrons (delta rays) of significant range and do not simultaneously impact the long cylindrical groups of cells traversed by high energy accelerated ions. Both neutron generating systems (RARAF accelerator) and Colorado State University 252Cf facility produce neutrons accompanied by a significant dose contribution from gamma rays which may ~20%. A few remarks about the differences in radiation types would be helpful and the fact that the uniformly distributed neutron energy deposition events are able to produce significant CNS responses is important in the sense that spatially correlated cell group damage may be only a minor contributor to the overall high LET effects. It also suggests that genotoxicity mechanisms may be drivers as the short recoil tracks on the scale of a cell nucleus may be indistinguishable from long tracks.
In lines 45 and 54, the term “α-particles” is used to indicated accelerated helium nuclei. This is a slang usage as α-particles are helium nuclei ejected as a result of radioactivity and never achieve energies > ~9 MeV (total, 2.25 MeV/n). 4He is properly used in line 474.
In line 59 it is indicated that neutrons comprise 10-30% of the dose inside spacecraft. This is somewhat misleading as levels of >=30% would occur only for impractically thick shielding situations. Norbury et al. (2019) show that at the most likely shielding level of 20 g/cm2 (most effective compromise between absorbing primary particles and generating secondary particles) the contributions is likely to be up to 12 – 15% for large solar particle events and up to 17 – 19% for galactic cosmic rays. The dose rate estimates for neutrons reported by Norbury et al. are from 25 to 35 mSv/yr in free space out of an estimated 900 day Mars mission dose equivalent of 1.2 Sv (about 3%) indicating that albedo neutrons from planetary surfaces and atmospheres are major contributors.
[John W. Norbury, Tony C. Slaba, Sukesh Aghara, Francis F. Badavi, Steve R. Blattnig, Martha S. Clowdsley, Lawrence H. Heilbronn, Kerry Lee, Khin M. Maung, Christopher J. Mertens, Jack Miller, Ryan B. Norman, Chris A. Sandridge, Robert Singleterry, Nikolai Sobolevsky, Jan L. Spangler, Lawrence W. Townsend, Charles M. Werneth, Kathryn Whitman, John W. Wilson, Sharon Xiaojing Xu, Cary Zeitlin.
Advances in space radiation physics and transport at NASA.
Life Sciences in Space Research. Volume 22, 2019, Pages 98-124, ISSN 2214-5524.
https://doi.org/10.1016/j.lssr.2019.07.003.]
Lines 106 – 128 discuss whole cell recordings and find no significant differences in intrinsic cell/membrane properties but altered action potential frequencies. Significant changes in resting membrane potential, input resistance and connection probability are observed with charged particles in the same dose range with high energy protons as well as heavier ions. The low energy recoil proton vs high energy (>= 150 MeV) protons are therefore different in a way related to the track structure, not the macroscopic absorbed dose.
In the Light-Dark test the number of transitions were reported with no observed changes. Were other parameters changed such as relative time spent in the light or dark zones or distances traveled there?
In the Forced Swim Test it is often observed that during the first couple of minutes, all animals move and struggle and that in the last minute or so most display helplessness and are immobile. Thus many authors report %time immobile for the middle period of the test where effects of treatment diverge. Please clarify whether % time immobile was for the full test period or a selected time period.
In the discussion, references to intellectual impairments in atomic bomb survivors may not be accurate. Impairments are associated with exposures to children with developing brains and higher levels of neurogenesis than adults. Vascular dementias are somewhat elevated in the overall atomic bomb cohort but not Alzheimers or Parkinsons.
In the discussion lines 393 – 407 suggest that altered neuron morphology may be responsible for reducing excitability (sEPSC frequency). Isn’t a significant GABAergic mechanism also likely? For example, in dentate gyrus with increased inhibitory tone (vs CA1) radiation effects on LTP are generally smaller. And, in line 172 it is mentioned that elevated inhibitory signaling is seen in hippocampus after charged particle exposures but results with sIPSCs and neutrons show no changes. It is suggested that homeostatic compensation and/or dysregulation may be causing some observed behavioral effects despite the lack of electrophysiological changes. This is an attractive explanation and GABA receptors, metabotropic ion channels and trophic factors such as BDNF may play roles in this regard but were not examined in the present study.
The idea that outcome measures linked to more complex circuitry are likely to be more radiation sensitive because there are more things that can go wrong is attractive and is consistent with observations that more complex behaviors are more radiosensitive.
This acute exposure study paired with the previous chronic neutron study in principle provide an opportunity to estimate dose rate effectiveness factors which are important correction factors in risk assessment. Do the reported results lend themselves to such estimates?
Editorial issues
Typos at lines 301 (neuron should be neutron) and 633 (than should be then)
Line 309 refers to “anxiety-like” behaviors versus “distress” behaviors on line 72. Anxiety-like is probably preferred.
Author Response
Thank you for your comments - we have posted our response to all reviews in the attached file

Reviewer 2 Report
The study entitled “Acute, low-dose neutron exposures adversely impact nervous system function“ is very interesting and may provide us with applicatory insight into health effects of neutron irradiation. The study uses range of properly chosen methods, it is clearly written and easy to follow. However some issues need to be clarified and corrected before further proceeding.
First of all, the authors should provide rationale for choosing exactly the dose of 18cGy, and secondly, how the time-frames were chosen. Why the experiments were performed 3-5 months after irradiation? Aren’t the effects time-dependent?
Should not the authors perform dose and time-response studies first?
Line 276: In Fear extinction experiment please explain why 30cGy was used? “0 cGy and 30cGy, respectively”
In the same experiment what does the numbers after “F” mean? i.e. F(2,32)
What does it mean “N=17/13 animals for NOR”?
Author Response

(The authors gave the same response as above.)

Reviewer 3 Report
This is an interesting article which demonstrates the effects of acute neutrons on nervous system function of mice. Their observations are based on in vivo and ex vivo study, by testing electrophysiology and mice behavior, the authors have suggested the neurological risk of acute neutron exposure. My specific concerns are outlined below.
While the study is timely and carefully designed, the title and description should be more matching what was actually measures. Nervous system function can be analyzed at multiple levels.
Because the behavioral experiments were carried out over a long period of time, it might be informative to illustrate as a general figure for readers to understand the paradigm of this experiment more easily.
Description of mice irradiation is brief. There is no details about dosimetry. Dosimetry detail is very important. Authors must provide dosimetry details under method section.
Approval number is needed regarding IACUC. In addition, how many animals were used for in this study?
Reference style should be modified to fit the journal format (Numbered).
Author Response

(The authors gave the same response as above.)

Reviewer 4 Report
The manuscript by Klein and colleagues describes the effects of acute neutron exposure (18 cGy) on several different behavioral endpoints, in addition to measuring excitatory and inhibitory synaptic currents, and LTP within the CA1 region of the hippocampus. Overall, the paper is well-written, the study is well-designed and controlled, and provides interesting dose rate comparisons to previous work for similar CNS endpoints from the same group. The lack of LTP changes is an important distinction between this acute neutron exposure and an earlier paper published by this group examining chronic neutron exposure (18cGy). These data suggest that while acute exposure studies provide some information about the deleterious effects of radiation exposure on the brain, chronic studies are needed to fully understand how the brain is affected by HZE exposure. Overall, acute neutron exposure still induced behavioral deficits in multiple cognitive domains and decreased excitatory synaptic signaling within the hippocampus.
The lack of assessments in any region other than the hippocampus is a weakness of the paper. Many of the behavioral tasks the authors employed involve activity within other brain regions, including the perirhinal cortex, amygdala, medial prefrontal cortex, etc., but these regions were not investigated in the current study. Further, the authors have previously reported differences within these other brain regions and have correlated these changes with behavioral performances, primarily in the OiP task. Given the interesting, and potentially important, differences between acute and chronic neutron exposures, the lack of these studies or studies of additional targets within the hippocampus, limits the mechanistic insights provided by the current study. The authors provide several readily testable hypotheses about what could underlie these differences within the CA1, for example, but none are tested in the current manuscript.
Minor comments:
- For some of the behavioral tests the authors denote when in time they occur (e.g., OiP was tested one week after NOR), but these details are not included for every group of behavioral tests. The authors should either add a timeline figure or simply add these details to the text (preferred). It’s assumed that social interaction testing occurred at 2 months following exposure, but how long after social interaction testing was the NOR run? How long after OiP were the emotionality tests completed? How long after this was fear conditioning/extinction completed?
- While the methods state a total of 31 mice were used for the current study (control = 17; neutron irradiated = 14), please clarify why subject numbers differ in the figure captions. For example, some behavioral tests have n=13 or n=16. For the ephys studies, please clarify total number of mice used (it appears to be 11?).
Author Response

(The authors gave the same response as above.)

Round 2
Reviewer 2 Report
The authors adequatly responded to my critiques. My recommendation is to accept the paper